# Analysis of the Gut Microbiome and Dietary Habits in Metastatic Melanoma Patients with a Complete and Sustained Response to Immunotherapy

**DOI:** 10.3390/cancers15113052

**Published:** 2023-06-04

**Authors:** Marin Golčić, Luka Simetić, Davorin Herceg, Krešimir Blažičević, Gordana Kenđel Jovanović, Ivan Dražić, Andrej Belančić, Nataša Skočibušić, Dora Palčevski, Igor Rubinić, Vera Vlahović-Palčevski, Tea Majnarić, Renata Dobrila-Dintinjana, Stjepko Pleština

**Affiliations:** 1Department of Radiotherapy and Oncology, University Hospital Center Rijeka, 51000 Rijeka, Croatia; marin.golcic@gmail.com (M.G.); renatadobrila@windowslive.com (R.D.-D.); 2Department of Oncology, University Hospital Center Zagreb, 10000 Zagreb, Croatia; 3Department of Health Ecology, Teaching Institute of Public Health of Primorsko-Goranska County, 51000 Rijeka, Croatia; 4Department of Mathematics, Physics and Foreign Languages, Faculty of Engineering, University of Rijeka, 51000 Rijeka, Croatia; 5Department of Clinical Pharmacology, University Hospital Center Rijeka, 51000 Rijeka, Croatianatasa.skocibusic@gmail.com (N.S.); vera.vlahovicpalcevski@gmail.com (V.V.-P.); 6Department of Internal Medicine, University Hospital Center Rijeka, 51000 Rijeka, Croatia; dora.palcevski@gmail.com; 7Community Health Center of Primorsko-Goranska County, 51000 Rijeka, Croatia

**Keywords:** dietary habits, gut microbiome, immunotherapy, melanoma, response time

## Abstract

**Simple Summary:**

Immunotherapy is the basis for treating metastatic melanoma. However, most patients will not achieve complete remission. While we have yet to determine which patients will respond to immunotherapy, a growing body of evidence emphasizes the role of gut microbiome and diet. Our study wanted to evaluate whether metastatic melanoma patients with a complete and sustained response to immunotherapy, which was previously thought to be a homogeneous group, exhibited different dietary habits and gut microbiome based on the time required to achieve a response. We showed that patients who exhibited complete remission after more than 9 months since the start of immunotherapy reported a significantly lower intake of proteins and sweets and a higher intake of flavones. They also exhibited a particular microbiome profile, previously associated with an improved response to immunotherapy. These results suggested that particular microbiome and diet are associated with a late and sustained response to immunotherapy.

**Abstract:**

Immunotherapy has improved the prognosis of metastatic melanoma patients, although most patients do not achieve a complete response. While specific gut microbiome and dietary habits might influence treatment success, there is a lack of concordance between the studies, potentially due to dichotomizing patients only into responders and non-responders. The aim of this study was to elucidate whether metastatic melanoma patients with complete and sustained response to immunotherapy exhibit differences in gut microbiome composition among themselves, and whether those differences were associated with specific dietary habits. Shotgun metagenomic sequencing revealed that patients who exhibited a complete response after more than 9 months of treatment (late responders) exhibited a significantly higher beta-diversity (*p* = 0.02), with a higher abundance of *Coprococcus comes* (LDA 3.548, *p* = 0.010), *Bifidobacterium pseudocatenulatum* (LDA 3.392, *p* = 0.024), and lower abundance of *Prevotellaceae* (*p* = 0.04) compared to early responders. Furthermore, late responders exhibited a different diet profile, with a significantly lower intake of proteins and sweets and a higher intake of flavones (*p* < 0.05). The research showed that metastatic melanoma patients with a complete and sustained response to immunotherapy were a heterogeneous group. Patients with a late complete response exhibited microbiome and dietary habits which were previously associated with an improved response to immunotherapy.

## 1. Introduction

The prognosis of metastatic melanoma has been dramatically improved in recent years due to the development of checkpoint inhibitors, anti-programmed cell death-1, anti-cytotoxic T-lymphocyte-associated protein 4, and anti-lymphocyte-activation gene 3 immunotherapy. While a median overall survival of 72 months was recently reported for dual immunotherapy, two-thirds of patients progressed after 6.5 years of follow-up [1,2,3].

Various patient characteristics and dietary habits might influence treatment success in a metastatic melanoma setting. The use of both probiotics [4] and antibiotics [5] was associated with a worse response to checkpoint inhibitors, while high-salt [6] and high-fiber diets [4,7] were linked with an improved response to immunotherapy. The development of 16s rRNA and metagenomic sequencing tools has also elucidated the role of the fecal microbiome in immunotherapy response. However, there is a lack of concordance in defining beneficial and detrimental bacterial taxa in metastatic melanoma patients [4,6,7,8,9,10,11,12,13,14,15,16,17,18]. Generally, Actinobacteria and Firmicutes phyla were associated with a good response, while Bacteroidetes and Proteobacteria phyla were linked with a poor response to immunotherapy [16,19,20]. However, the majority of research on this topic dichotomized patients to responder vs. non-responder cohorts, evaluating responders as a homogenous group.

Despite a lack of consensus on the optimal microbiome, research on both mice [7,8,11] and human patients [17,18] have shown that fecal microbial transplantation (FMT) taken from responders can improve the response to immunotherapy and achieve a clinical benefit in progressing patients with metastatic melanoma. Although all donors were required to have a significant and sustained response to immunotherapy, the clinical response of recipients differed dramatically based on the donors [17,18]. While the discrepancies between the responses were not completely understood, recent studies have shown that the baseline gut microbiome might be primarily responsible for the late response to immunotherapy in metastatic melanoma patients [15,16]. Hence, there could be, microbiome-wise, at least two different categories of responders to immunotherapy, which could partially explain the inconsistent results previous research. Furthermore, despite the significant association between diet, microbiome, and response to immunotherapy [4,6,7], there is a scarcity of research evaluating both factors together.

The aim of this study was to elucidate whether metastatic melanoma patients with a complete and sustained response to immunotherapy exhibit differences in gut microbiome composition among themselves, and whether those differences were associated with specific dietary and lifestyle habits.

## 2. Materials and Methods

### 2.1. General Study Design and Participants

The observational, cross-sectional study was performed at the Clinical Hospital Center Zagreb, Croatia, and included metastatic melanoma patients treated with immunotherapy between April 2017 and October 2020, who experienced a complete and sustained response longer than 12 months (N = 15). The response was confirmed by at least two positron emission tomography/computerized tomography (PET/CT) scans, while values of S100 and LDH proteins were required to be within referral ranges. All patients who participated in this study were over 18 years old, had a histopathologically confirmed melanoma, and had a baseline PET/CT. The patients received immunotherapy as the first or second line of treatment. Immunotherapy protocols included: (1) pembrolizumab 200 mg IV every 3 weeks, (2) nivolumab 240 mg IV every 2 weeks, or (3) the combination of nivolumab 1 mg/kg + ipilimumab 3 mg/kg every 3 weeks, followed by nivolumab 240 mg IV every 2 weeks after 4 cycles. The diagnostic follow-up was performed every 12 weeks as per local recommendations. The patients were required to have at least 5 cycles of immunotherapy and at least 2 PET/CTs. iRECIST criteria were used for radiological evaluation. Pseudoprogression was defined as an initial increase in tumor size followed by a decrease in tumor burden on further evaluations. LDH and S100 biomarkers were routinely evaluated. Patients’ basic demographic information and information regarding the disease, treatment, and laboratory results were independently derived from electronic medical records by two oncology specialists. Patients were excluded from the study if they did not satisfy the inclusion criteria or if they reported recent use of probiotics or antibiotics.

### 2.2. Fecal Microbiome Analysis

After potential candidates with proper inclusion criteria were identified in a cross-sectional manner, patient stool samples were collected using OMNIgene OM200 kits with DNA stabilization (DNA Genotek, Ottawa, ON, Canada). The kits were used according to the manufacturer’s instruction and immediately frozen in a −80 °C monitored freezer before being transported to CosmosID in May 2022 (Rockville, MD, USA), where DNA extraction and additional analyses were performed. 

DNA from samples was isolated using the QIAGEN DNeasy PowerSoil Pro Kit according to the manufacturer’s protocol. Extracted DNA samples were quantified using the GloMax Plate Reader System (Promega, Madison, WI, USA) using the QuantiFluor dsDNA System (Promega) chemistry. DNA libraries were prepared using the Nextera XT DNA Library Preparation Kit (Illumina, San Diego, CA, USA) and IDT Unique Dual Indexes with total DNA input of 1 ng. Genomic DNA was fragmented using a proportional amount of Illumina Nextera XT fragmentation enzyme. Unique dual indexes were added to each sample, followed by 12 cycles of PCR to construct libraries. DNA libraries were purified using AMpure magnetic Beads (Beckman Coulter, Brea, CA, USA) and eluted in QIAGEN EB buffer. DNA libraries were quantified using a Qubit 4 fluorometer and Qubit™ dsDNA HS Assay Kit. Libraries were then sequenced on an Illumina NextSeq 2000 platform 2 × 150 b.

Bioinformatics analysis was also performed by CosmosID (Rockville, MD, USA), with a system utilizing a high-performance data-mining k-mer algorithm that rapidly disambiguates millions of short sequence reads into the discrete genomes engendering the particular sequences. The pipeline had two separable comparators: the first consisted of a pre-computation phase for reference databases, and the second was a per-sample computation. The inputs for the pre-computation phase were databases of reference genomes, virulence markers, and antimicrobial resistance markers. The output of the pre-computational phase was a phylogeny tree of microbes, together with sets of variable-length k-mer fingerprints (biomarkers) uniquely associated with distinct branches and leaves of the tree. The second per-sample computational phase searched the hundreds of millions of short sequence reads, or alternatively contigs from draft de novo assemblies, against the fingerprint sets. This query enabled the sensitive yet highly precise detection and taxonomic classification of microbial NGS reads. The resulting statistics were analyzed to return the fine-grain taxonomic and relative abundance estimates for the microbial NGS datasets. To exclude false positive identifications, the results were filtered using a filtering threshold derived based on internal statistical scores that were determined by analyzing a large number of diverse metagenomes. The same approach was applied to enable the sensitive and accurate detection of genetic markers for virulence and for resistance to antibiotics.

### 2.3. Evaluation of Dietary and Lifestyle Habits

Dietary intake and lifestyle habits were noted through telephone calls by instructed interviewers during May and June 2022. This part of analysis coincided with the stool sample collection. Physical activity habits were assessed with the short-form International Physical Activity Questionnaire (IPAQ) [21], while dietary habits were evaluated by four separate detailed 24 h food recalls over a period of three weeks from the first interview, on non-consecutive days and on different days of the week, as previously reported [22,23]. Patients’ diets were separated into different dietary components for further analysis. The patients’ dietary energy and nutrient intakes were calculated by using the Croatian food composition database [24], while for the intakes of certain nutrients, including caffeine, β-carotene, omega-3 and omega-6 fatty acids, and phenolic compounds, the Danish [25] and American food composition database [26] and the Phenol-Explorer 3.0 database were used [27].

The daily resting energy expenditure was calculated for each patient with Mifflin–St. Jeor’s equation [28], using their age, gender, and anthropometric data, which was then multiplied by the activity factor based on information from the IPAQ. Patients’ diet quality was assessed by comparing their average energy and nutrient intakes with their estimated adequate intakes based on their age, gender, and daily resting energy expenditure [29].

Patients’ diet was further evaluated with the MDS for adherence to the MD [30]. The MDS is based on 9 characteristics of the traditional MD. The total MDS ranges from 0 to 9; an MDS of 0–3 indicates no adherence to the MD, 4–5 indicates medium adherence, and 6–9 good adherence to the MD. 

For assessing the inflammatory potential of the diet, the Dietary Inflammatory Index (DII^®^) was used [31]. The DII^®^ is calculated from energy and nutrient intakes obtained from patients’ 24 h recalls. Those intakes were firstly linked to the global means and standard deviations of the food and nutrient intakes from 11 nations to calculate the z-scores and were then converted to a percentile and centered to minimize the “right skew” by doubling the value and subtracting 1. The provided percentile score of each nutrient was multiplied by the respective inflammatory effect score to deliver the food-parameter-specific DII score. The overall DII score of each patient’s diet comprised of a sum of forty-five food-parameter-specific DII scores. The study used 37 food parameters, which included nine pro-inflammatory components (energy, carbohydrates, protein, total fat, saturated fatty acids, trans fat, cholesterol, iron and vitamin B12) and 28 anti-inflammatory components (monounsaturated fatty acids; polyunsaturated fatty acids; omega-3 fatty acids; omega-6 fatty acids; fiber; alcohol; vitamins A, D, E, C, and B6; b-carotene; thiamine; riboflavin; niacin; folic acid; magnesium; selenium; zinc; flavan-3-ol; flavones; flavonols; flavonones; anthocyanidins; caffeine; garlic; onion; and pepper). The positive values of the DII score point to a pro-inflammatory diet, and the negative DII score values to an anti-inflammatory diet [31].

### 2.4. Statistical Analysis

Both alpha (diversity within the sample) and beta diversity (diversity between the samples) were evaluated using the online CosmosID Hub (Rockville, MD, USA). Alpha diversity was calculated using the Wilcoxon Rank-Sum Test (Chao1 and Simpson indexes), while beta diversity was calculated using permutational multivariate analysis of variance, evaluating both Jaccard distance and Bray–Curtis dissimilarity between the groups. Additional ordination was performed using the Principal Coordinate Analysis. Diversity analyses were performed for different patient characteristics. All further analyses in the manuscript primarily evaluated the differences between metastatic melanoma patients with early and late complete response, as a time to complete the response that was longer or shorter than 9 months was the lone patient characteristic exhibiting a statistically significant difference in beta diversity analysis. 

A linear discriminant analysis (LDA) effect size (LEfSE) was performed to evaluate the difference in the relative abundance of bacterial species between the patient cohorts, using a statistical hub provided by CosmosID, with the threshold of LDA = 3 (Rockville, MD, USA). 

The differences in abundance in phylum and family levels were analyzed only for taxa reported by previous studies to have either positive or negative associations with the success of immunotherapy treatment as quoted in the main document. The analyses were performed using both relative proportion of bacterial species and the abundance score, an absolute normalized abundance metric taking into consideration genome size and number of reads, provided by the CosmosID software (CosmosID-HUB v2.0) (Rockville, MD, USA). The Mann–Whitney test was used to analyze the difference between the abundance of bacterial groups, while repeated measures analysis of variance was used to analyze the difference in food intake between subjects and within subjects. In cases where the estimated sphericity was greater than 0.75, the Huynh–Feldt correction was used to analyze the within-subject effects; otherwise, the Greenhouse–Geisser correction was used. Chi-squared test, *t*-test, and Mann–Whitney tests were also used to compare the differences between patient characteristics, depending on the type of data.

Pearson correlations were performed to evaluate the relationship between the different dietary and lifestyle categories and specific bacterial taxa previously reported to have an association with either improved or worsened response to immunotherapy, using both absolute scores and the relative percentages of bacteria. The correlations were performed on phylum, family, and species levels. All calculations were performed using MedCalc Statistical Software version 14.8.1, Ostend, Belgium. Because the statistical sample was relatively small, the threshold of weak statistical significance (*p* < 0.1) was considered satisfactory as an indicator of the association between dietary habits and response to immunotherapy, in addition to the standard threshold of statistical significance (*p* < 0.05) [32].

## 3. Results

### 3.1. Basic Characteristics

A total of 15 patients were included in the study. The majority of patients were male (N = 12, 80.0%), non-smokers (N = 13, 86.7%), and retired (N = 12, 80.0%), with an average age of 61.0 (±12.2) years. Patients reported regular use of prescription drugs (N = 13, 86.7%), most commonly blood-pressure medications (N = 9, 60.0%), but did not report recent use of probiotics or antibiotics. 

Regarding the melanoma, most patients were BRAF-negative (N = 10, 66.7%), with the most common initial stages being T4 (N = 6, 40%) and N1 (N = 7, 46.7%). Upon the diagnosis of metastatic disease, most commonly in the lungs (N = 9, 60.0%), the majority of patients received mono-immunotherapy (N = 12, 80.0%) in the first-line setting (N = 14, 93.3%). Two patients (13.3%) exhibited signs of pseudo-progression. A total of five patients (33.3%) required more than 9 months to achieve CR. Treatment was well tolerated with transient side-effects in the majority of patients, with five patients (33.3%) requiring systemic corticosteroids at any time point, and two patients (13.3%) receiving long-term systemic therapy. No patients reported significant gastrointestinal autoimmune side-effects. Additional details regarding the disease and biomarkers are given in Table 1. 

### 3.2. Diversity Analysis

The difference in alpha diversity was found for the number of metastatic sites, with patients who had three or more metastatic sites exhibiting higher alpha diversity compared to patients with a lower burden of disease (Chao1 1.97 (*p* = 0.05), Simpson 2.08 (*p* = 0.04)) (Appendix A).

Beta diversity analysis revealed a statistically significant difference only based on the time required for a complete response to immunotherapy treatment. Patients who required more than 9 months to achieve CR (classified further in the text as the late responders) exhibited higher beta diversity (*p* = 0.02) compared to early responders (Figure 1 and Appendix A). There were no differences in beta diversity based on the time required for partial response (*p* = 0.57 on Bray–Curtis, *p* = 0.89 on Jaccard). 

There were no differences between the early and late responders in any of the analyzed patient characteristics (Appendix A), while full analysis of the differences in alpha and beta diversity for all analyzed patient characteristics are presented in Appendix A.

### 3.3. Abundance of Gut Bacteria

Late responders exhibited higher abundance of Coprococcus comes (linear discriminant analysis (LDA) 3.548, *p* = 0.010) and Bifidobacterium pseudocatenulatum (LDA 3.392, *p* = 0.024) along with unspecified Bacteria, Bifidobacteria, and Eggerthella (LDA < 3, *p* < 0.05). Barnesiella intestinihominis (LDA 3.582, *p* = 0.036), Sutterella wadsworthensis (LDA 3.386, *p* = 0.038), and Bacteroides finegoldii (LDA 3.051, *p* = 0.02) were more commonly found in the early responder group (Figure 2). 

On the bacterial family level, a significantly lower number of Prevotellaceae was found in late responders (*p* = 0.046), with a trend toward a higher number of *Bifidobacteriaceae*, *Clostridiales*, *Lactobacillaceae*, *Ruminococcaceae*, and *Lachnospiraceae*, compared to early responders. The opposite results were found for *Bacteroidaceae*, *Akkermansiaceae*, and *Coriobacteriaceae* (Table 2). Similar data were found for the differences in relative abundance of the same bacterial families (Appendix A).

Regarding the bacterial phyla, a numerically higher number of Actinobacteria and Firmicutes and a lower number of Bacteroidetes and Proteobacteria species were registered in late responders (*p* > 0.05) (Appendix A). Similar results were found for the differences in relative abundance (Appendix A). The Firmicutes-to-Bacteroidetes ratio was 2.2 in early responders and 3.4 in late responders, although the difference was not significant (*p* > 0.05). 

No differences in alpha diversity, beta diversity, or abundance scores for virulence factors and antimicrobial resistance analysis between the early and late responders were recorded.

### 3.4. Dietary Characteristics

The average body mass index (BMI) of all patients was 27.02 (±3.52) kg/m^2^, with five patients (33.3%) exhibiting a normal BMI. The average dietary energy intake was within individual recommendations (98.2% (±12.4%)), and the mean metabolic equivalent of task (MET) was 2753.9 (±1215.7) min/day. Two-thirds of patients (66.7%) were considered at least moderately active based on their average weekly physical activity habits (Appendix A). 

The majority of patients (N = 13; 86.7%) reported habitual consumption of home-grown food, predominantly vegetables. Regular intake of industrially processed food was reported by six patients (40%), while only one (6.7%) patient consumed artificial sweeteners. The mean inflammatory potential of the diet, assessed with dietary inflammatory index (DII^®^), was 1.55 (±1.76), characterizing a diet with a pro-inflammatory potential, with no statistically significant differences between the late and early responders (DII 0.69 ± 2.36 vs. 1.98 ± 1.92, *p* = 0.19). The Mediterranean Diet (MD) Score (MDS) for the whole group of patients was 5.90 (±1.34), which is viewed as a medium-to-good adherence to MD, with no significant differences between the groups (MDS 6.40 ± 2.06 vs. 5.65 ± 1.81; *p* = 0.167) (Appendix A). 

A detailed comparison between the dietary habits of the two patient cohorts is presented in Appendix A, while a summary of dietary categories with either statistical significance (*p* < 0.05) or weak significance (*p* < 0.1) between the groups is displayed in Table 3. 

Late responders consumed higher amounts of flavones (9.1 vs. 3.9 mg/day; *p* = 0.027), but less proteins (recommended protein intake per body weight (133.9 vs. 178.1%; *p* = 0.005)) and sweets (14.6 vs. 54.2 g/day; *p* = 0.04), compared to early responders. Differences with weak statistical significance (*p* = 0.05–0.09) were registered for vitamin D, vegetable, anthocyanin, alcohol, and saturated fatty acid use between the two groups (Table 3).

No differences in dietary fiber (25 vs. 22 g/day; *p* = 0.54) or sodium intake (3.6 vs. 3.8 g/day; *p* = 0.65) were noted between the group in between the groups, even though all patients consumed more than the recommended values of sodium (Appendix A) [33]. 

According to univariate logistic regression analysis, which is fully presented in Appendix A, there was a significant association between alcohol, potato, polyunsaturated fatty acid, protein, sweets, vegetable, vitamin D, and saturated fatty acid intake and the time to complete response to immunotherapy.

### 3.5. Correlations of Dietary Habits and Bacterial Taxa

A detailed analysis of correlations of dietary habits and bacterial taxa is given in Appendix A. On the phylum level, significant correlations using both absolute scores (AS) and relative percentage (RP) were found for Firmicutes and the consumption of anthocyanins (r = 0.56, *p* = 0.03 (RP), and r = 0.52, *p* = 0.04 (AS)) (Figure 3). In contrast, a significant correlation between Actinobacteria and consumption of all vegetables was found using RP (r = 0.67, *p* < 0.01), but not when analyzing the AS (r = 0.50, *p* = 0.06 (AS)). 

For the bacterial family level, the abundance of the *Bacteroidaceae* correlated with both energy share of PUFA (r = 0.53, *p* = 0.04, AS and RP) and the intake of vitamin D (r = 0.69, *p* < 0.01 for both AS and RP). Additionally, flavone intake correlated with *Lactobacillaceae* (r = 0.54, *p* = 0.04 (AS); r = 0.53, *p* = 0.04 (RP)) and *Ruminococcaceae* (r = 0.56, *p* = 0.03 for both AS and RP). A similar correlation was observed between the consumption of alcohol and both *Lactobacillaceae* (r = 0.64, *p* = 0.01 (AS), r = 0.67 (RP)) and *Ruminococcaceae* (r = 0.79, *p* < 0.01 (AS), r = 0.70, *p* < 0.01 (RP)).

Regarding the species level, a strong and significant correlation was found between Bifidobacterium pseudocatenulatum and the vitamin D intake (r = 0.74, *p* < 0.01 (AS), r = 0.53 (*p* = 0.04) (RS).

Various bacterial phyla, families, and species exhibited a moderate correlation to different dietary components but with weak statistical significance (*p* = 0.05–0.09) (Appendix A). 

Consumption of both saturated fatty acids and fibers exhibited no significant correlations to any bacterial phylum, family, or species analyzed. Likewise, Proteobacteria phylum, Bifidobacteria, Clostridiales, and Akkermansiaceae families, and Barnesiella intestinihominis species, exhibited no significant correlations with any food components, although the level of activity (MET min/day) correlated with both Proteobacteria (r = 0.56, *p* = 0.03) and Bacteroidetes phyla (r = 0.55, *p* = 0.03) and Barnesiella intestinihominis species (r = 0.59, *p* = 0.02) similarly on AS and RP analysis.

## 4. Discussion

Although there is a growing body of evidence showing the importance of the gut microbiome in cancer treatment, elucidating the importance of specific bacteria has been challenging [4,6,8,9,10,11,12,13,14,15,16,17,18]. The differences in DNA extraction methods, bioinformatic pipelines, and sequencing platforms could be partially responsible for the discordance of the results [34,35]. However, while the majority of research dichotomized the patients into responder and non-responder cohorts [4,6,7,9,11,12,13,14,15,16], recent data have shown that the baseline microbiome primarily affected the late response to immunotherapy, while host-intrinsic and tumor-intrinsic factors were suggested as responsible for early responses [15,16].

Furthermore, while various dietary and lifestyle habits, including the use of medications, dietary supplements, and the level of exercise, were found to have a tremendous impact on the gut microbiome [4,5,7,36,37,38], the majority of trials on this topic, including the pivotal FMT trials [17,18], did not evaluate dietary habits along with the gut microbiome analysis. 

Hence, we decided to perform the combined analyses in patients with a complete and sustained response to immunotherapy, which were previously evaluated as a homogeneous group. A shotgun metagenomic sequencing of the gut microbiome was preferred to 16 s rRNA sequencing, due to its potential to analyze all genomic data in a sample [39]. Since recent studies downplayed the value of alpha diversity (diversity within the sample) in response to immunotherapy [17,18], this study was primarily focused on beta diversity, a measure that represents a difference between the two microbial communities. 

The results showed a significant difference between the gut microbiome of early and late complete responders to immunotherapy, with a cut-off of 9 months to complete response, which is in line with previous research [15,16]. 

Analysis of the abundance of bacterial taxa showed that two bacterial species with the highest LDA in the late-response group were *Coprococcus comes* and *Bifidobacterium pseudocatenulatum*, which both belong to a phylum and family previously associated with a positive effect on the immunotherapy response [16,17,18]. Furthermore, *Coprococcus comes* was shown to produce butyrate from fiber [40], which was associated with a significant response to immunotherapy [19,41] and was associated with a higher lymphocyte count in patients with SARS-CoV-2 infection [42]. *Bifidobacterium pseudocatenulatum* was already linked to an excellent response to immunotherapy in melanoma patients [15], along with an improvement in neural and immune function and inflammatory status [43,44]. On the family and phylum level, only *Prevotellaceae*, previously shown to be enriched in melanoma patients who did not show substantial response to immunotherapy [16], had a significantly lower expression in the late responders.

The three bacteria with the highest LDA in early responders were *Barnesiella intestinihominis* and *Bacteroides finegoldii*, species of the Bacteroidetes phylum, along with *Sutterella wadsworthensis*, a member of the Proteobacteria phylum. Both phyla were previously associated with a worse response to immunotherapy in melanoma patients [16,19]. However, *Barnesiella intestinihominis* was actually linked with good outcomes in renal cancer treatment [45], while *Bacteroides finegoldii* was generally a marker of a healthy gut but was not reported in studies with cancer patients [46]. On the contrary, *Sutterella wadsworthensis* was associated with gastrointestinal diseases [47].

While an earlier study defined a beneficial Firmicutes-to-Bacteroidetes ratio for immunotherapy response between 0.5–1.5 [48], this study reported an average ratio of 2.6 (3.4 in late responders), with only 3 out of 15 (20%) patients exhibiting the aforementioned ratio. Since Baruch et al. reported that a 60% response rate was achieved with FMT using a donor with a Firmicutes-to-Bacteroidetes ratio of 23.9, a higher ratio might be valuable in melanoma patients treated with immunotherapy [18].

Although two-thirds of our patients were obese or overweight, elevated BMI was previously associated with improved outcomes in immune checkpoint inhibitor treatment [49,50]. Patients exhibited a diet that moderately adhered to MD, although with a mild pro-inflammatory effect. Most patients (80%) consumed >20 g of fiber a day, compared to 29% in the trial by Spencer et al., which showed the potential benefit of dietary fiber during immunotherapy treatment [4]. Furthermore, all patients in this study consumed more than the recommended daily sodium intake (2.0 g/day), which could be beneficial, since a high-salt diet was associated with an improved immunotherapy response [6,33,51]. 

A comparison between the two cohorts showed that the late responders consumed significantly higher amounts of flavones, a subgroup of flavonoids, which have been shown as an effective sensitizer for anti-cancer therapy partially by modulation of the immune response [52]. Furthermore, flavones have been shown to increase the population of butyrate-producing species such as *Ruminococcus* and *Coprococcus* [53], which were associated with an improved response to immunotherapy [41]. On the other hand, lower consumption of proteins in late responders might also be beneficial since a protein-restricted diet was linked with a more robust response to immunotherapy, possibly through the inhibition of the mTOR pathway [54]. Late responders also consumed sweets in lower amounts, which was considered a healthy habit due to the many metabolic detrimental effects linked to overconsumption of sweets, an association with melanoma incidence [55], and a shift toward a more pro-inflammatory microbiome [51].

Late responders also reported a trend toward higher values of PUFA energy intake share, vitamin D and anthocyanin consumption, as well as lower reported intake of saturated fatty acid use compared to early responders, which was all previously associated with potentially beneficial effects on the cancer treatment [51,55,56,57,58,59,60]. A higher use of alcohol was reported in late responders; while chronic alcoholism was linked with the development of various cancers, a potential of alcohol to activate dendritic and natural killer cells and potentially increase PD-L1 protein expression in tumor tissue was also reported [61,62]. However, the alcohol intake was within guidelines for recommended alcohol consumption [63]. 

The study also found several significant positive correlations between the consumption of specific dietary categories and different bacterial taxa, including the abundance of Firmicutes phylum and anthocyanin intake. Although previous studies have recorded divergent results [64], the different proportion of specific bacterial families within the phylum could explain this inconsistency, as anthocyanin supplementation was actually shown to increase the abundance of *Lachnospiraceae*, a member of Firmicutes phylum [65]. On the family level, the abundance of *Bacteroidaceae* correlated with both PUFA (in line with work by Patterson et al.) [66] and vitamin D intake. The abundance of *Lactobacillaceae* and *Ruminococcaceae* were associated with intakes of both flavones, as previously shown [53,67,68], and alcohol, possibly due to the consumption of wine, which is rich in flavones and was the most common alcoholic drink consumed by the study patients [69]. Finally, on the species level, a strong positive correlation between *Bifidobacterium pseudocatenulatum* and vitamin D intake was found, although there was no correlation with the *Bifidobacteria* family as a whole. Physical activity level, which could potentially enhance the immunotherapy treatment, positively correlated with both Proteobacteria and Bacteroidetes phyla, and *Barnesiella intestinihominis* species, partially in line with previous but scarce research on the topic [70,71]. In summary, higher consumption of anthocyanins, flavones, PUFA, vitamin D, and alcohol, which were all consumed in higher amounts in the late responder group, positively correlated with bacterial phyla, families, and species previously associated with a more robust response to immunotherapy [4,10,11,15,16,19].

## 5. Conclusions

Although the majority of previous studies on this topic have evaluated metastatic melanoma patients with a response to immunotherapy as a single group, our results demonstrated the presence of significant heterogeneity in both microbiome and dietary habits even among the patients with a complete and sustained response, based on the time to response [4,6,7,9,11,12,13,14,15,16].

Metastatic melanoma patients with a late complete response exhibited higher beta diversity, higher abundance of *Coprococcus comes* and *Bifidobacterium pseudocatenulatum*, and lower abundance of *Prevotellaceae* compared to early responders. While both groups of patients had an elevated BMI, consumed dietary fibers in desirable quantities, and had higher than recommended sodium intake, late responders also consumed less sweets and protein, and more flavones, compared to early responders; all of which was previously shown to have the potential to enhance a response to immunotherapy. Furthermore, the study found positive correlations between specific dietary components associated with the MD [72] and Firmicutes phylum, *Bacteroidaceae*, *Lactobacillaceae*, and *Ruminococcaceae* families, and *Bifidobacterium pseudocatenulatum* species.

This study adds to the relatively scarce literature on the gut microbiome and dietary habits in melanoma patients treated with immunotherapy, showing that any future evaluations must take into consideration the heterogeneity among the responders. 

## 6. Limitations

Several limitations in this research needed to be acknowledged. Firstly, this was a study with a cross-sectional design, which did not enable us to assess the causality, but only the association. Future studies should answer the question whether particular dietary interventions or selection of FMT donors based on time to response could result in the optimization of the treatment of melanoma patients. 

Furthermore, baseline microbiome analysis was not performed, only the analysis during the period of sustained complete response, which limited us in evaluating microbiome stability and difference at various time points. However, there was an abundance of data showing relative microbiome stability over time, especially if no antibiotics or probiotics were used [11,16,17]. 

Additionally, potential FMT donors would generally be required to exhibit a period of prolonged and sustained response to immunotherapy, but would rarely have a baseline microbiome analysis. A relatively small number of studied patients could be considered as a study limitation, but was comparable to the largest trial in the same field which included a total of 22 complete responders [15]. We do recommend confirming the results in a larger cohort of patients.

Secondly, 24 h recall interviews to evaluate dietary habits were chosen rather than the Food Frequency Questionnaire (FFQ), commonly used in diet–disease studies. Using FFQ could have resulted in a systematic and random error when assessing dietary intake, which could have affected estimates of diet–disease associations [73]. Additionally, since the information of repeated 24 h dietary recalls was documented on non-consecutive days, the probability of daily food item consumption could be estimated more accurately compared to the FFQ [74]. Although trained staff performed the interviews, a possible recall bias still existed, and while dietary habits were evaluated during the period of complete response, they were not evaluated during the active medical treatment. 

Thirdly, the results were derived from a homogeneous population of Caucasian Croatian patients with high level of home-grown food intake, which might not be generalized, although the patients lived in different regions of Croatia and accurately represented an average population of Croatian melanoma patients. There is also a potential bias in recollection of use of probiotics or antibiotics by the patients. In future studies, it is recommended to include the control group of melanoma patients with a poor response to immunotherapy.

## Figures and Tables

**Figure 1 cancers-15-03052-f001:**
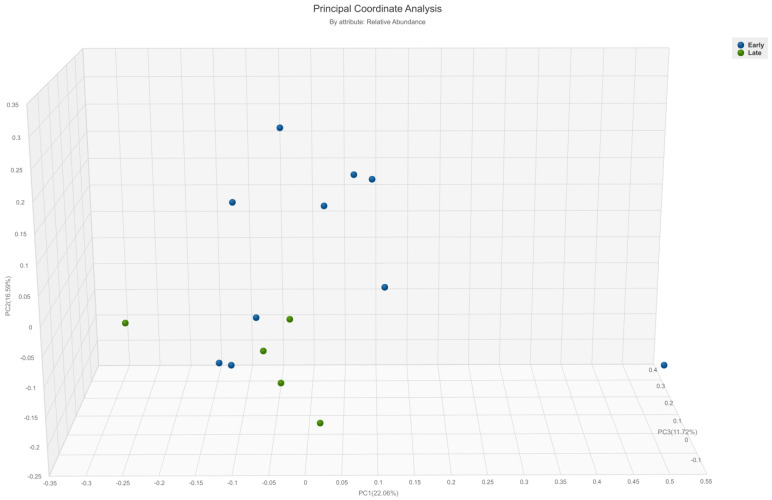
A principal coordinate analysis graph (Bray–Curtis) showing the difference in beta-diversity between metastatic melanoma patients who required more or less than 9 months to achieve complete response on immunotherapy (*p* = 0.02). Green = late responders, blue = early responders.

**Figure 2 cancers-15-03052-f002:**
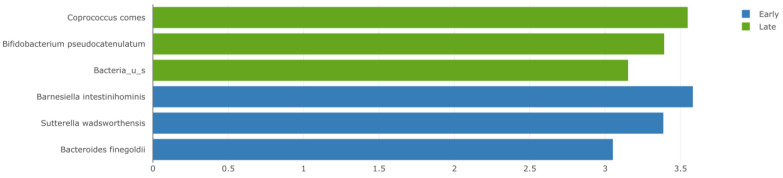
A linear discriminant analysis (LDA) effect size (LEfSE) plot showing the bacterial strains with the difference in relative abundance between metastatic melanoma patients who exhibited either early or late response to immunotherapy. Green = late responders, blue = early responders. LDA threshold = 3, *p* < 0.05.

**Figure 3 cancers-15-03052-f003:**
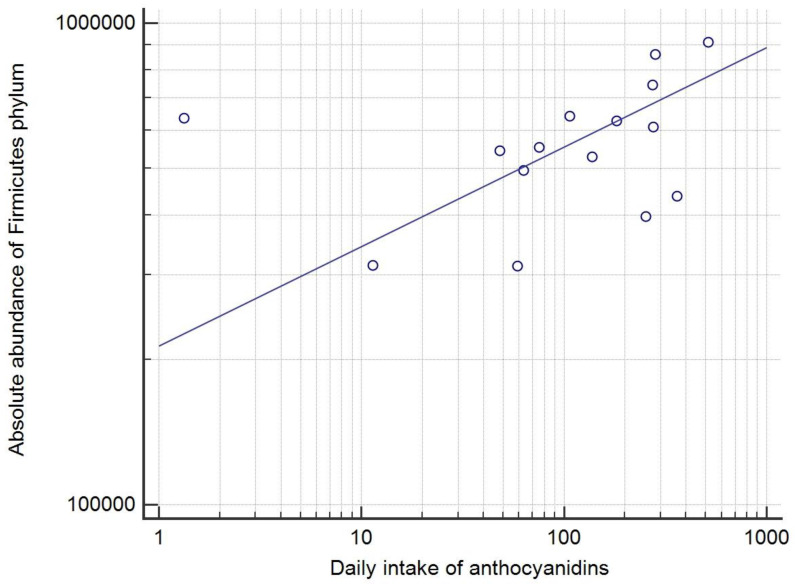
The correlation between the dietary anthocyanin intake (mg/day) and the absolute abundance score of Firmicutes phylum (r = 0.52, *p* = 0.04), with the logarithmic transformation of both x and y axis and a trend line. Each circle represents a single patient.

**Table 1 cancers-15-03052-t001:** Descriptive statistics of the continuous variables in the whole patient population.

Characteristics	Median	Mean	St. Dev	Range
Age (years)	65.0	61.0	12.2	31.6–74.6
Time from local to metastatic disease (years)	1.3	1.5	1.0	0.3–3.3
Number of metastatic sites (number of sites involved)	2.0	2.3	1.1	1.0–4.0
Time to any response (months)	3.4	4.1	2.1	2.6–10.6
Time to complete response (months)	6.7	7.6	4.6	2.6–16.0
Length of complete response (months)	42.0	39.1	15.6	12.0–62.0
Initial LDH (U/L)	178.0	229.8	78.7	161.0–370.0
Final LDH (U/L)	174.0	188.5	39.7	128.0–251.0
Initial S100 (μg/L)	0.1	0.3	0.6	0.0–0.3
Final S100 (μg/L)	0.0	0.0	0.0	0.0–0.1

**Table 2 cancers-15-03052-t002:** The difference in abundance score of specific bacterial families previously associated with a significant effect on immunotherapy in melanoma patients between early and late metastatic melanoma responders to immunotherapy.

Family	Early Responders (N = 10) ^a^		Late Responders (N = 5) ^a^		*p*-Value
	Median	25–75 Percentile	Median	25–75 Percentile	
*Akkermansiaceae* [19]	955.845	0.000–6226.710	217.010	134.820–3533.513	0.49
*Bacteroidaceae* [19]	92,336.290	65,543.770–111,755.850	108,741.230	70,196.573–169,369.500	0.54
*Bifidobacteriaceae* [9,17]	15,790.415	5811.140–26,845.590	28,342.210	16,359.345–31,951.560	0.33
*Clostridiales* [18]	18,207.495	11,291.520–22,377.090	26,986.570	17,743.852–31,394.840	0.22
*Coriobacteriaceae* [17]	4504.415	1977.590–6574.440	2063.890	1635.570–4188.727	0.27
*Lachnospiraceae* [16,17]	263,232.330	171,838.330–369,730.370	330,307.830	234,938.785–368,924.418	0.62
*Lactobacillaceae* [19]	6626.785	4967.030–10,770.830	9630.150	1266.413–17,395.037	0.62
*Prevotellaceae* [11,19]	**10,175.440**	**79.360–58,930.220**	**0.000**	**0.000–3614.740**	**0.046**
*Ruminococcceae* [4,11,17]	103,276.270	98,685.480–115,307.020	138,048.170	110,200.172–219,744.712	0.08

^a^ The cut-off for late responders was 9 or more months to reach complete response. Bolded values denote statistical significance (*p* < 0.05).

**Table 3 cancers-15-03052-t003:** The difference in mean intake of selected dietary components (*p* < 0.1) between metastatic melanoma patients with early and late response to immunotherapy.

	Early Responders (N = 10) ^a^		Late Responders (N = 5) ^a^		Between-Subjects Effect	Within-Subjects Effect
Dietary Component	Mean	SD	Mean	SD		
Alcohol (g/day)	25,536	636,602	118,572	1,904,176	0.090	0.517 *
Anthocyanin (mg/day)	131,328	2,094,609	266,063	3,336,942	0.094	0.340 **
Flavones (mg/day)	**3616**	**39,908**	**9073**	**91,212**	**0.027**	**0.640 ***
Potatoes (g/day)	103,036	1,368,100	188,286	1,422,395	0.067	0.540 *
Polyunsaturated fatty acids (% energy intake)	5818	24,680	8386	42,103	0.099	0.541 *
Proteins (% recommended protein (g) use per body weight (kg))	**178,100**	**505,530**	**133,946**	**231,728**	**0.005**	**0.365 ****
Sweets (g/day)	**54,170**	**569,123**	**14,590**	**234,487**	**0.040**	**0.664 ****
All vegetables (g/day)	375,706	2,192,604	560,321	2,953,988	0.051	0.740 *
Vitamin D (mcg/day)	2079	22,959	6380	78,023	0.050	0.416 *
Saturated fatty acids (% energy intake)	16,160	60,581	12,030	42,199	0.058	0.299 **
Saturated fatty acids (g/day)	36,627	139,657	27,927	116,296	0.058	0.284 **

^a^ The cut-off for late responders was 9 or more months to reach complete response. Bolded values denote *p* < 0.05. Detailed daily dietary components are given in Appendix A. * Huynh–Feldt ** Greenhouse–Geisser.

## Data Availability

Raw data were generated with CosmosID (Rockville, MD, USA). Microbiome data supporting the findings of this study were deposited with the NCBI and are available at: https://www.ncbi.nlm.nih.gov/sra/PRJNA915098 (accessed on 23 December 2022). The repository with dietary data is available at the Mendeley Data Repository: https://doi.org/10.17632/h9kk9cgjzk.1 (accessed on 23 December 2022).

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
