# Peer review of "Analysis of the Gut Microbiome and Dietary Habits in Metastatic Melanoma Patients with a Complete and Sustained Response to Immunotherapy"

_cancers, 2023, doi:10.3390/cancers15113052_

Round 1

Reviewer 1 Report

‘Metastatic melanoma patients with a late complete response to immunotherapy exhibit different microbiome and dietary habits compared to early responders’ provided interesting insights into the differences in microbiome and dietary habits between early and late responders to immunotherapy in metastatic melanoma patients. However, there are still some aspects that need improvement.

Major comments:

·    The study on the microbiome and dietary habits of metastatic melanoma patients receiving immunotherapy is interesting, but there is an important issue that needs clarification. Specifically, it is unclear if the patient stool samples collection time and the assessment of dietary and lifestyle habits were consistent. The authors should provide more information on this matter.

·    In addition, as diet has a significant impact on both the microbiome and disease outcomes, it is difficult to establish a causal relationship between microbiome differences and response to therapy based on a single experiment. Therefore, the observed microbiome differences alone may not be sufficient to conclude that they are a significant factor affecting the response of metastatic melanoma patients to immunotherapy.

·    I recommend adding a table containing basic characteristics of the study population under the 3.1 results section. This would help readers to better understand the patient population and the distribution of basic characteristics between the late responders and early responders groups. Additionally, I suggest performing statistical tests to compare the two groups, for categorical variables such as gender and smoking status using chi-squared tests, for continuous variables such as age using Wilcoxon Rank Sum tests.

·    I have some concerns regarding the data analysis and interpretation. Specifically, I feel that the data analysis and mining may not be comprehensive enough to support the conclusions made in the study. The study employed basic analysis, but I feel that more advanced statistical techniques such as regression analysis and machine learning algorithms could have been used to uncover more insights from the data. Additionally, the sample size is relatively small, and the two groups are not evenly distributed, which could affect the statistical power of the results.

Minor comments:

·    I have noticed that in the manuscript, the names of microorganisms are not italicized, such as Bifidobacteriaceae, Akkermansiaceae and so on. As scientific writing convention, it is necessary to italicize the names of microorganisms to distinguish them from regular text. Therefore, I recommend the authors to revise the manuscript accordingly.

Author Response

Please see the attachment; our response was written in bolded blue color. 

Reviewer 2 Report

[General Comments]

This is a small study that tries to understand an interesting question about the difference in microbiome and dietary habits in early vs late responders of immunotherapy. The authors can be applauded for their detailed analyses of the gut microbiome and dietary habits. However, the actual significance of their finding and how it can help advance the field is not as strong. 

The main criticism of the paper is that it may not matter whether a patient is an early or late responder. A complete response is a complete response. Unless the authors are able to show that late responses result in longer complete responses or improved overall survival (in their paper or citing other papers), the difference in their gut bacteria and dietary habits may not ultimately matter. The authors state in their conclusion that "patients with the late response to immunotherapy might be more suitable as donors for the future FMT trials." They are again basing this on the fact that late responders exhibited microbiome, dietary, and lifestyle habits which were previously shown to have potential to enhance a response to immunotherapy. However again, if they cannot show that late responses are better than early responses I do not agree with this statement. Similarly, the early responders exhibited microbiome/dietary habits which have previously been shown to be associated with a worse response to immunotherapy, but these patients DID have a response to immunotherapy. How do the authors explain this finding? 

The generalizability of this study is also quite limited as it was mostly males getting monotherapy whom many ate home-grown foods (86.7%); this would certainly not be as relevant to a population such as the US where many female and male patients get combination therapy and do not eat such a diet.

As the authors stated, the paper could be strengthened if the authors included information on poor responders and how this may be different to early/late responders. The paper could also be strengthened if the patients had baseline stool sampling and dietary recalls as well as during and after treatment. Finally, an interesting question that could add to the paper is whether any of these patients had GI immunotherapy side effects such as colitis, and whether that impacted their microbiome and response.

[Detailed Comments]

Introduction: The first sentence states that the prognosis of metastatic melanoma has been improved by recent immunotherapy development. I would include LAG-3 inhibitors here. 

Materials and Methods: Please include timing of diet recalls and stool samples. Was this the same timing for all patients? Was it collected once a patient was determined to be in complete response or any time once a complete response was determined?

Supplemental figure 4: The legend cuts off the groups "Ear...ers" and "Late...ers"

Author Response

(The authors gave the same response as above.)

Round 2

Reviewer 1 Report

I would like to express my appreciation to the authors for their careful consideration of my previous comments for the manuscript titled "Analysis of the gut microbiome and dietary habits in metastatic melanoma patients with a complete and sustained response to immunotherapy."

The revised manuscript demonstrates significant enhancements in the content. These improvements have substantially strengthened the overall presentation and understanding of the study.